



# Evaluation of uncertainties in mean and extreme precipitation under climate changes for northwestern Mediterranean watersheds from high-resolution Med and Euro-CORDEX ensembles

Colmet-Daage Antoine[1,2,3], Sanchez-Gomez Emilia[1], Ricci Sophie[1], Llovel Cécile[2], Borrell Estupina Valérie[3], Quintana-Seguí Pere[4], Llasat Maria Carmen[5], Servat Eric[6]

[1]CECI, CERFACS – CNRS TOULOUSE, Toulouse, France
[2]WSP France, Toulouse, France
[3]Hydrosciences Montpellier, Univ. Montpellier, Montpellier, France
[4]Observatori de l'Ebre Fundació, Tarragona, Spain
[5]Universitat de Barcelona, Barcelona, Spain
[6]Institut Montpelliérain de l'Eau et de l'Environnement – IRD, Montpellier, France

*Correspondence to*: Colmet-Daage Antoine (colmet@cerfacs.fr)

**Abstract.** The climate change impact on mean and extreme precipitation events in the northern Mediterranean region is assessed over high resolution EuroCORDEX and MedCORDEX simulations. The focus is made on three regions, the Lez and the Aude located in France, and the Muga, located in northeastern Spain and eight pairs of global and regional climate models are analyzed with respect to the SAFRAN product. First the model skills are evaluated in terms of bias for the precipitation annual cycle over past period. Then future changes in extreme precipitation, under two emission scenarios, are estimated through the computation of past/future change coefficients of quantile-ranked model precipitation outputs. Over past period, the cumulative precipitation is overestimated for most models over the mountainous regions and underestimated over the coastal regions in autumn and higher order quantile. The ensemble mean and the spread for future period remain unchanged under RCP4.5 scenario and decrease under RCP8.5 scenario. Extreme precipitation events are intensified over the three catchments with a smaller ensemble spread under RCP8.5 revealing more evident changes, especially in the last part of the 21th century.

## 1 Introduction

IPCC SREX report (IPCC 2012) concludes with an increase in the frequency of heavy precipitation episodes over most areas of the globe at the end of 21st century. In particular, Mediterranean northwestern regions are often affected by extreme precipitation events that generate flash floods and cause serious damages (Ricard et al., 2012; Gaume et al., 2016). This climatically homogeneous region (Metzger et al., 2005) has been identified as a hot spot of climate changes (CC) in the form of possible amplification of extreme precipitation associated with a decrease in total precipitation (Gao et al., 2006; Giorgi, 2006; Giorgi and Lionello, 2008; Milano et al., 2013). Assessing the impacts of regional climate change on precipitation



constitutes a major challenge, in order to help and support policy makers to develop strategies facing to future hydrological vulnerabilities like flash floods.

General climate models (GCMs) are powerful tools to assess global scale climate variability and change. GCMs have allowed a better understanding of interactions between the different components of the climate system (atmosphere, ocean, sea-ice, continents). However GCMs generally operate at coarse horizontal resolutions (100-250 km in the atmospheric component), hence they are not appropriate to investigate the hydrological impacts of future extreme precipitation events at local scale, as over hydrological watersheds that are as small as hundreds of kilometers.

The interest of better representing climate variability and change at local scale motivated the development of regional climate models (RCMs), which currently are able to perform dynamical downscaling of GCM at very high horizontal resolutions (~ 10 km). RCMs run on limited area domains thereby allowing increased spatial resolution, and thus enabling a better representation of surface heterogeneities and mesoscale atmospheric processes like convection (Fowler et al., 2007a). At European scale, collaborative research projects such as MERCURE (Hagemann et al., 2004), PRUDENCE (Christensen et al., 2007), NARCCAP (Paulsen et al., 2009) and ENSEMBLES (van der Linden and Mitchell, 2009) have contributed to further developments and improvements in regional modeling. More recently, the international CORDEX (Coordinated Regional Climate Downscaling Experiment) initiative (Giorgi et al., 2009) has provided multi-model regional climate simulations at very high spatial resolution over different regions in the world. In particular, for the northwestern Mediterranean region, the EuroCORDEX and MedCORDEX subprojects (EMCORDEX hereinafter) have produced present and future climate simulations at 12 km resolution.

The merits of increased spatial resolution in RCMs models have largely been assessed in the literature. Comprehensive evaluations of RCMs have been undertaken over the Euro-Mediterranean region by applying evaluation metrics to mean values of precipitation (Déqué and Somot, 2010; Fisher et al., 2012; Jacob et al., 2007; Kjellström et al., 2010; Kotlarski et al., 2005) as well as focusing on extreme precipitation associated with hydrological floods (Frei et al., 2006; Fowler et al., 2007b; Herrera et al., 2010; Kyselỳ et al., 2012; Maraun et al., 2012). For recent EMCORDEX models, initial evaluations over past periods have been conducted over Europe (Drobinski et al., 2016; Katragkou et al., 2015; Kotlarski et al., 2014). The latter evaluations mainly focused on mean and extreme precipitation over the whole EMCORDEX domain or in large regional boxes (e.g. France, The Alps, Mediterranean coastal regions, Morocco) using sparse observed datasets. Prein et al. (2016) highlighted the added value of high-resolution models (12.5 km versus 50 km) for the simulated mean and extreme precipitation thanks to an improved representation of orography and large scale convection. Indeed, Drobinski et al. (2016) show the higher ability of high resolution RCMs to reproduce the Clausius-Clapeyron relation and thus precipitation-related processes. In a climate change context, Jacob et al. (2013) show that future climate projections performed by high-resolution RCMs (12.5km) scenario under the Representative Concentration Pathways 4.5 and 8.5 (RCP4.5 and RCP8.5 respectively) project higher daily precipitation intensities than GCMs, in particular for RCP8.5. These results are consistent with Giorgi et al. (2016) conclusions over the Alps region. Even though both GCMs and RCMs scenario experiments project a reduction of



summer precipitation over the Alps Mountains, increased convective rainfall due to enhanced potential instability related to a finer representation of the orography over the Alps region is found in RCMs.

Together with increasing model resolution, high-resolution observation-based products have also been recently developed over different European and Mediterranean countries. Moving on from CRU product at 50 km resolution (Harris et al., 2014), to E-OBS product at 25 km resolution (Haylock et al., 2008), reanalysis such as SAFRAN (Système d'Analyse Fournissant des Renseignements Atmosphériques à la Neige; Durand et al., 1993; Quintana-Seguí et al., 2008) or interpolated products such as SPAIN02 (Herrera et al., 2012, 2016) now provide precipitation products at a resolution comparable to RCMs'. In particular, the SAFRAN dataset available over France (Quintana-Seguí et al., 2008, Vidal et al., 2010) and Spain (Quintana-Seguí et al., 2016a, 2016b) comprises a much larger observed data network than E-Obs or ERA-Interim (Dee et al., 2011) that were previously used for EMCORDEX models assessment (Cavicchia et al., 2016; Kotlarski et al., 2014). The SAFRAN-France dataset was used together with downscaled products issued from CMIP5 (Coupled Model Intercomparison Phase 5) to assess future hydrological changes over France (Dayon, 2015; Quintana-Seguí et al., 2010, 2011). Themeßl et al. (2011) provides a review of downscaling methods. This implies a systematic bias correction of model precipitation before being used as input for hydrological models, for instance in the framework of future flash floods simulation. Harader (2015) used the regional model ALADIN5.2 outputs at 12km resolution from CORDEX as well as SAFRAN-France product to describe future flash flood events over the Lez catchment using a "futurization" method described in the following.

The present study focuses on extreme precipitation over mesoscale northwestern Mediterranean watersheds with a complex orography. Three watersheds of various sizes are investigated here: the Lez and the Aude located in southern France, and the Muga, located in northeastern Spain. The goals of this study are:

• To assess RCM skills from the EMCORDEX multi-model ensemble in terms of mean and extreme precipitation values over past periods.

• To assess the influence of GCMs lateral boundaries condition on the RCMs skills.

• To evaluate future changes in precipitation extremes for further simulations of flash floods with an event-based hydrological model over future periods.

For this purpose, the "futurization" approach proposed by Harader (2015) is used. This method stands in the computation of a past/future change coefficients of quantile-ranked RCM precipitation outputs. Each multiplicative coefficient is then applied to each quantile-ranked short-term observed precipitation event. It should be noted that the quantile-ranked observed precipitation are computed from SAFRAN daily precipitation that have generated flash floods. The futurization method is applied for each RCM in the EMCORDEX ensembles, forced by two emission scenarios (RCP4.5 and RCP8.5). We thus explicitly take into account climate model-related uncertainty. In further work, the "futurized" precipitation events will be used with different hydrological models, so that we would explicitly take into account hydrology model-related uncertainty.

The paper is organized as follows, section 2 includes a brief presentation of the EMCORDEX simulations, the reference datasets and the statistical metrics applied to seasonal mean and extreme precipitation values. Section 3 presents the RCM



evaluation in terms of mean and extreme precipitation values over the present period when the global scale is prescribed by ERA-Interim. Section 4 analyzes present climate simulations to understand the role of the GCMs in driving the RCMs. The impacts of climate change on precipitation are then examined in Section 5. Conclusion and perspectives are finally given in section 6.

## 5 2 Data and Methods

### 2.1 The River Catchments

In the current study, the "futurization" approach is applied over three Mediterranean watersheds with different characteristics and external influences. The Lez, Aude and Muga catchments displayed in Fig. 1, are frequently subject to flash floods that cause considerable damages to surrounding areas and cities.

The upstream part of the Lez watershed, framed in red in Fig. 1, is located 15 km north of the city of Montpellier and covers 114 km². The landscape is dominated by garrigue vegetation, very common in the Mediterranean countries. The spring of the Lez River is the resurgence of a karstic aquifer of about 380 km². The karst aquifer plays an important role in the water resource of the basin, and the karsts outcrops actively participate in flash floods dynamics (Raynaud et al. submitted). Cumulated annual rainfall is around 909 mm which, on average, falls on 60 days per year (Coustau, 2011; Harader, 2015).

The Lez catchment is frequently subject to flash floods caused by extreme precipitation episodes regionally known as the « Cévenols » events (Ducrocq et al., 2008; Nuissier et al., 2008, 2011). Cumulated extreme precipitation can locally reach 600 mm in 24 h within the river catchment (Boudevillain et al., 2011).

The Aude watershed, framed in brown in Fig. 1, covers more than 5,000 km² in the upstream neighboring of Narbonne city. The Aude River is born at the Pyrenees and flows along the catchment for 223 km before entering the Mediterranean Sea.

The Orbieu and the Fresquel are its major tributaries. The Aude catchment is surrounded by several mountain chains as the Cevennes massif to the north and the Pyrenees to the south. The catchment is mainly under the influence of a Mediterranean climate, but large climate contrasts can be found over its sub-watersheds. A severe flash flood episode occurred in November 1999 (more than 200 mm of rain in 24h over a major part of the catchment), and caused severe damage over an extended region (Aude, Tarn, Pyrénées Orientales, Hérault) (Estupina, 2004; Bechtold & Bazille, 2001; Ducrocq et al., 2003; Gaume

et al., 2004).

Finally, the Muga catchment (framed in purple in Fig. 1) located in northeastern Spain over the Catalonia, Region covers 854 km². The Muga River is about 58 km long, between the Pyrenees (maximum altitude of the catchment, 1214 m) and the Gulf of Rosas. This catchment is usually affected by heavy precipitations associated to convective events, with an annual precipitation average between 700 mm in the upper part and 530 in the mouth, and a daily precipitation of 200 mm for return

period of 10 years (Llasat et al., 2009, 2014). The previously mentioned November 1999 flash flood event led to the maximum historical peak discharge ever recorded near the mouth of the Muga River, with 925 m³ s⁻¹ against an average





value of 3.34 m$^3$ s$^{-1}$. The Muga catchment was affected by 26 severe flood events between 1982 and 2010 (Llasat et al., 2014).

It should be noted that, in this paper, statistical analysis are carried out over the Lez, Aude and Muga catchment areas as well as over more extended regional boxes also displayed in Fig. 1.

## 2.2 RCM Euro-Mediterranean CORDEX simulations

The set of EMCORDEX simulations used in this study are summarized in Tab. 1. The five used RCMs are presented with the main reference papers (in particular with respect to boundary layer and convection schemes). The EMCORDEX community has provided three types of RCM simulations with driving conditions (also detailed in Tab. 1) issued from either ERA-Interim or GCMs simulations over past and future periods:

- The Evaluation simulations (EVAL hereinafter). The lateral boundary conditions (LBCs) are driven by ERA-Interim reanalysis (Dee et al., 2011) over 1981-2010. These simulations are used to evaluate the RCM intrinsic biases.

- The Historical simulations (HIST hereinafter). The LBCs are issued from numerical experiments performed with four different GCMs and extracted from the CMIP5 historical archive. These historical simulations represent the climate conditions over 1976-2005.

- The Future Climate Scenario simulations (RCP hereinafter). In this case, the LBCs correspond to four future climate scenarios (from four different GCMs from CMIP5) under RCP4.5 and RCP8.5 (Clarke et al., 2007; Riahi et al., 2007; Meinshausen et al., 2011; van Vuuren et al., 2011) over 2011-2040, 2041-2070 and 2071-2100. These simulations cover 30-year time-slice periods like the HIST simulation. Thus they can be statistically compared to assess future changes.

This results in eight pairs of RCM-GCM simulations analyzed in this study, for HIST and RCP as indicated in the last row of Tab. 1. Only simulations at 12 km for which EVAL, HIST and both RCP experiments were available at the beginning of the study were considered.

## 2.3 The reference dataset : SAFRAN

SAFRAN re-analysis provides daily precipitation data for the period 1958-2008 over France and Spain on an 8 km and 5 km grid respectively (Vidal et al., 2010, Quintana-Seguí et al., 2016a, 2016b). SAFRAN France was built by using data from 3 675 selected rain gauges that were gridded trough an optimal interpolation algorithm described in Quintana-Seguí et al. (2008). The great number of rain gauges considered in SAFRAN and its high spatial resolution produce more accurate precipitation analyses over France and Spain than those proposed by other products commonly used for model assessments, such as CRU (Harris et al., 2014) and E-OBS (Haylock et al., 2008). A recent study shows that, for precipitation, the performance of SAFRAN is very similar to that of Spain02 (Quintana-Seguí et al. 2016b). SAFRAN dataset has been





evaluated using the Météo-France and Spanish State Meteorological Agency (AEMET) gauging station network as well as independent data (Quintana-Seguí et al. 2008, 2016a, 2016b; Vidal et al. 2010).

SAFRAN product is used here as a reference dataset to evaluate the simulated precipitation from EVAL and HIST ensembles. For that purpose, the 12 km RCMs outputs and also SAFRAN Spain were regridded on the 8 km SAFRAN-France grid using the ESMF (Earth System Modeling Framework) regridding bilinear method. Complementary tests have shown that the interpolation does not induce any significant error in the precipitation fields.

## 2.4 Statistical metrics to evaluate RCM performance

Given the small size of the Lez, Aude and Muga catchments (3, 84 and 11 SAFRAN grid points respectively) and in order to allow for a proper statistical analysis, the evaluation of mean precipitation (seasonal and annual cycle) was achieved on larger regional boxes (excluding sea grid points) shown in Fig. 1. These regional boxes have been selected according to regions of homogeneous climate conditions. In the particular case of the Lez catchment, the RCM precipitation is evaluated over the entire Cevennes region. However for all three catchments, extreme precipitation metrics are computed over grid points that are strictly inside the catchments. The precipitation extremes are analyzed with respect to the 90[th] to 99.9[th] quantiles of the daily precipitation distribution discretized as follows:

- One point per 1 quantile rank from 90[th] to 95[th];
- One point per 0.5 quantile rank from 95[th] to 98[th];
- One point per 0.2 quantile from 98[th] to 99[th];
- One point per 0.1 quantile from 99[th] to 99.9[th].

Short-term observed precipitation events are thereby completely covered and evenly distributed with this quantile discretization (not shown). Quantiles are computed considering all the days (rainy days and dry days), thus allowing for a comparison of precipitation quantiles between the different RCM-GCM pairs (Giorgi et al., 2016; Schär et al., 2016). Yet, it should be noted that when precipitation is below 0.1 mm day$^{-1}$, the precipitation is set to 0 so that the cumulative precipitation is not affected by an over-representation of light rainy days in the models (Harader et al., 2015; Tramblay et al., 2013, Paxian et al., 2015).

In addition to the classical metrics (spatial bias, annual cycle bias, quantiles-quantiles plot), two original metrics are used in this study. First, assuming additivity between GCM and RCM errors, the impact of the GCM bias on the RCM solution can be diagnosed by computing the difference $\Delta B$ between the HIST and the EVAL precipitation bias with respect to SAFRAN:

$$\Delta B = \frac{HIST-SAFRAN}{SAFRAN} - \frac{EVAL-SAFRAN}{SAFRAN} \quad (1)$$

Secondly, in the present study, change coefficients between the past (HIST) and future precipitation (RCP) quantile distributions are computed. For that purpose, HIST and RCPs precipitations are quantile-ranked, and for each quantile-rank "qi", a change coefficient "Aqi" for precipitation intensities between HIST and RCP is computed as follows:





$$Aqi = \frac{Pqi(RCP)}{Pqi(HIST)} \quad (2)$$

where *Pqi(RCP)* and *Pqi(HIST)* represent the values of the quantiles for HIST and RCP respectively.

All metrics are computed on a seasonal basis, considering hereinafter four seasons: autumn (September, October and November), winter (December, January and February), spring (March, April and May) and summer (June, July and August). Only autumn and spring results are presented in the following section.

## 3 Analysis of RCM Evaluation simulations

In this section the RCMs precipitation biases are diagnosed through the comparison between the EVAL simulations and SAFRAN. The spatial pattern for the mean cumulative precipitation, the annual cycle and the extreme values are investigated.

### 3.1 Spatial bias pattern

Figure 1 shows the spatial distribution of the cumulative precipitation normalized difference between each RCM from the EVAL ensemble and SAFRAN, averaged over the 30-year past period in autumn. The top-left panel in Fig. 1-a displays the mean cumulative precipitation for SAFRAN-France and SAFRAN-Spain reference datasets. Large values of the cumulative precipitation (superior to 400 mm season$^{-1}$) are observed over mountainous regions, in particular the Cevennes and the Pyrenees chains. Lower cumulative precipitation values are observed in the valleys (Garonne, Aude) and over the coastal regions.

In general, the cumulative precipitation is overestimated for most RCMs over the mountainous regions (+30%) and underestimated over the Mediterranean coastal region (-30%), as shown in Fig. 1-b to f, most likely because of an imperfect representation of the orography as well as the parameterization of the convection scheme. Note that the RACMO22E pattern of precipitation bias differs from the other RCMs patterns. Indeed, a slight overestimation is observed over the south of the Pyrenees and the Cevennes (+20%), and almost no bias is observed over the valleys.

We focus now on the precipitation bias over river catchments that are analyzed in the present study: For the Cevennes, the cumulative precipitation is overestimated by 20% in the southwest mountainous region and underestimated by 30% in the northeast valley region for all RCMs (Fig. 1-b-c-e-f) except for RACMO22E (Fig. 1-d).

For the Aude region, no relevant bias is represented by ALADIN52 (Fig. 1-b), ALADIN53 (c) and RACMO22E (d). A positive bias (+40%) is represented by RCA4 (Fig. 1-e) in the western region under continental influence, while no relevant bias is represented in the eastern region under Mediterranean influence. HIRHAM5 (Fig. 1-f) displays a strong positive bias (+50%) in the high elevation areas (Pyrenees in the southwest, and Black mountain in the northwest) and a strong negative bias elsewhere.





The mean cumulative precipitation is underestimated (-30%) by all RCMs over the Muga region, except for RACMO22E. For other seasons, the mean cumulative precipitation pattern tends to be overestimated over all three regions in spring (not shown) and a strong positive bias (+50%) is represented by ALADIN52 and ALADIN53 over the Pyrenees in summer (not shown).

## 3.2 Annual cycle of precipitation bias

The 30-year climatology for the monthly cumulative precipitation is spatially averaged over each region of interest and normalized by the SAFRAN monthly climatology. Figure 2-a displays the annual cycle of precipitation over the Aude box for SAFRAN dataset. Figure 2-b displays the normalized annual cycle over the Aude region for each RCM. Globally

speaking, the RCMs bias and the inter-model spread are smaller in autumn than in summer. This can be explained since the influence of the large-scale atmospheric circulation is weaker in summer, the control exerted by the LBCs on the RCM is reduced in this season, and the RCMs solution has more degrees of freedom to deviate from the large-scale forcing (Déqué et al, 2012; Lucas-Picher et al, 2008).

ALADIN52 and ALADIN53 clearly overestimate the cumulative monthly precipitation in late spring and summer. This is

15 likely due to a large presence of low precipitation days in the RCM precipitation distribution (0 to 10 mm day$^{-1}$) compared to SAFRAN distribution (not shown). This behavior is also observed over the Cevennes and Muga regions (not shown). Consistently with Fig. 1-d, RACMO22E displays a low bias in the annual cycle. While similar behavior is observed for HIRHAM5, this results in an error compensation between positive and negative spatial biases displayed in Fig. 1-f. For the Muga and the Cevennes boxes (not shown), HIRHAM5 presents a negative bias. Finally, RCA4 simulates a large positive

bias (+30%) for the entire annual cycle that tends to intensify over the Alps and the western Pyrenees.

## 3.3 Extreme precipitation bias

The ability of the EVAL simulations to represent the extreme of precipitation is investigated through a quantile-quantile analysis with respect to SAFRAN. This analysis is performed for each grid point within the Lez, Aude and Muga catchments. The highest order quantiles (95$^{th}$ to 99$^{th}$) are displayed in Fig. 3 for each RCM and SAFRAN in autumn.

For the Lez, Aude and Muga catchments, the RCMs underestimate the higher order quantiles. This underestimation is more important for the Muga catchment, where the extreme precipitation intensities are stronger.

ALADIN52 underestimates the upper precipitation quantiles, for instance, the intensity of the 99$^{th}$ quantiles only reaches 70 mm day$^{-1}$ against 140 mm day$^{-1}$ for SAFRAN. ALADIN53 performance is slightly improved with respect to ALADIN52, especially for the Aude and Muga catchments. RACMO22E and RCA4 underestimate the extreme precipitation above the

30 99.5$^{th}$ quantiles. HIRHAM5 provides a satisfying description of all extreme quantiles for the three catchments.





### 3.4 Summary of means and extremes analysis

The mean and extreme precipitation is investigated over the Aude, Lez and Muga catchments. Largest biases in mean precipitation simulated by the regional climate models are located over the mountainous (positive bias) and coastal (negative bias) regions. These biases are stronger during the summer season, when the control exerted by the LBCs on the RCM is

weaker, due to a reduction of the large-scale circulation and North Atlantic inflow. Overall, while the climate variability is covered by the spread of EMCORDEX ensemble of RCMs, each RCM simulates plausible precipitation.

These results are coherent with other studies for mean and extreme precipitation. ALADIN52 precipitation biases are consistent with those obtained by Harader, 2015, and slightly reduced in ALADIN53. The RCA4 overestimation of mean precipitation is in accordance with the results showed in Prein et al. (2016). The underestimation of extreme precipitation

diagnosed in our study is possibly even more severe, as our reference, SAFRAN, is itself deemed to underestimate observed extreme precipitation (Quintana-Seguí et al. 2008a). This last point was confirmed through a comparison between pluviometer data and SAFRAN over our catchments of interest.

Parameterization and physical processes that are involved in the generation of low precipitation differ from thus involved in stronger precipitation events. Thus the precipitation bias varies according to the precipitation intensity. A quantile

classification is thereby relevant for the formulation of a past/future precipitation change.

### 4 Analysis of RCM historical simulations

RCM intrinsic biases, determined from the EVAL ensemble, have been characterized and described in the previous section. Here we analyze the historical simulations (HIST), performed with the GCM-RCMs pairs, as described in section 2.2. In this case, GCM biases are expected to affect the RCM simulation. A strenuous question is to know how GCM and RCM biases

are interacting (Dequé et al. 2011). In this study we make the assumption of additivity between the impact of the GCM biases and the RCM intrinsic biases. In this study, 4 different GCM forcings are considered: CNRM-CM5, ICHEC, MOHC and MPI (see Table 1).

### 4.1 Annual cycle of precipitation bias

Figure 4 displays the annual cycle of ΔB over the Aude watershed for each GCM-RCMs pairs. The color code refers to the

GCM (for instance for CNRM-CM5 forcing), while the markers refer to the RCM (for instance star for ALADIN53). From Fig. 4 it is evident that CNRM-CM5 forcing leads to a systematic overestimation of summer precipitation, hence proving that the positive precipitation bias identified in EVAL (section 3, Fig. 2 for ALADIN52, ALADIN53 and RCA4) is enhanced in HIST simulations.

ICHEC large-scale conditions induce no significant changes on RACMO22E and RCA4 errors, while it leads to a strong

overestimation of the HIRHAM5 precipitation. Except when forced by CNRM-CM5, the bias of GCM-RCA4 pairs is similar



to the intrinsic bias of RCA4, as displayed for all three MPI, MOHC and HIRHAM5. Illustration is provided for the Aude box in Fig. 4 but similar results are obtained for the Cevennes and the Muga boxes.

It should be noted that previous results may be also explained by bias compensation between GCM impacts and RCM intrinsic bias. In despite of the GCM-RCM deficiencies shown here, all the GCM-RCMs pairs display plausible precipitation
and are then considered in the following for future change analysis over mean precipitation annual cycle.

## 4.2 Extreme precipitation bias

Figure 5 is the counterpart of Fig. 3; it displays quantile-quantile diagrams of precipitation in autumn for the HIST ensemble over the Lez, Aude and Muga catchments. First, the spread amongst the HIST simulations displayed in Fig. 5 is considerably larger than the one generated by the EVAL ensemble (Fig. 3) that the EVAL ones (Fig. 3), and extreme quantiles tend to be
systematically underestimated over all three catchments of interest.

Generally, CNRM-CM5 forcing leads to an underestimation of SAFRAN quantiles, thus enhancing the RCMs intrinsic biases displayed in Fig. 3. MPI and ICHEC provide good quantile distribution for all RCMs except beyond $99.5^{th}$ quantile. In more details, ICHEC-RCA4 extreme quantiles are slightly overestimated over the Aude catchment, and in good agreement with SAFRAN over the Muga catchment. MOHC-RCA4 provides good statistics over Aude and Muga
catchments, while it overestimates all quantiles over the Lez catchment.

To conclude, the impact of the GCM on RCMs tends to systematically intensify the underestimation of extreme precipitation values. Yet, HIST ensemble remains consistent with SAFRAN statistics and thus will be used in the following step, in order to estimate future changes over extreme precipitation over the different catchments.

## 5 Effect of climate change on precipitation

## 5.1 Annual cycle of precipitation

Figure 6 presents the precipitation annual cycle simulated for HIST computed over 1976-2005 period and the scenario RCP4.5 and RCP8.5 computed over 2071-2100 period, for the Cevennes, Aude and Muga boxes.

The results reveal a stronger mean precipitation change between RCP4.5 and RCP8.5 than for the 2011-2040 and 2041-2070 periods. The annual cycle of the RCP4.5 ensemble is similar to HIST one in terms of mean and spread, suggesting that
radiative forcing in RCP45 seem to induce a weak impact on monthly averaged precipitation. In contrast, the annual cycle of the RCP8.5 ensemble displays a general decrease in mean precipitation from April to October for the three river catchments. These results are consistent with the conclusions from the study of Jacob et al. 2013 over EMCORDEX domain. As previously mentioned the spread of the ensemble is larger in summer when the LBCs exert a weaker control in the RCM domain. It should be noted that these results hold for the three boxes of interest that are of significantly different sizes. This
advocates that the high resolution EMCORDEX simulations can be confidently used to investigate precipitation at local scale.





As mentioned in previous section, the analysis of the annual cycle of HIST simulations emphasized that the association of GCM and RCM models induces a different precipitation bias that the intrinsic RCM biases (comparison between black and blue curve in Fig. 6). Future precipitations from RCP simulations should thus be used after a bias correction step; formulated here thanks to change coefficients between past and future precipitation distribution.

## 5.2 Change coefficients for extreme precipitation

The change coefficients between quantile-ranked precipitations presented in Eq. 2 are displayed on Fig. 7. They allow estimating the changes between the past (HIST) and future precipitation (RCP) quantile distributions

Following equation 2, a change/transfer coefficient greater than 1 for a give quantile, indicates an increase of the future precipitation value associated with this quantile. On the other hand, $A_{qi} < 1$ means a decrease in RCP precipitation with respect to HIST.

The $A_{qi}$ coefficients are computed fo r each GCM-RCM pair over the Lez, Aude and Muga catchments. The multi-model approach adopted here allows estimating an uncertainty in the values of the change coefficients. The $A_{qi}$ coefficients are represented in Fig. 7 over the upper level quantile range [90-99.9] for RCP4.5 and RCP8.5 in autumn for the three river catchments and for 2041-2070 and 2071-2100 future time periods. The ensemble mean for $A_{qi}$ computed amongst the GCM-RCM pairs is compared to the associated ensemble spread (standard deviation) in Fig. 7.

The interesting results here is while RCP simulations tend to decrease mean precipitation with respect to HIST simulation (Fig. 6), extreme precipitation events are intensified for both time periods over the three catchments. Globally speaking, the mean change coefficient for RCP4.5 and RCP8.5 is similar except for the Lez (2041-2070) and the Muga (2071-2100) where RCP8.5 displays biggest changes. The ensemble spread from RCP8.5 is smaller than the one from RCP4.5, meaning that the change in extreme precipitation displays higher level of certainty under RCP8.5 scenario, which can be explained by a stronger radiative forcing in RCP85 compared to RCP45. Few models indicate a decrease in extreme precipitation for RCP4.5 over 2041-2070 period. In contrast, for 2071-2100, all the GCM-RCM members agree on an increase in extreme precipitation for both RCP4.5 and RCP8.5 scenarios. For instance, the mean change coefficient over 2071-2100 reaches 1.15 for the 99.5[th] quantile over the Aude catchment for both RCPs, while RCP4.5 spread is about 0.3 against 0.1 for RCP8.5. In terms of precipitation, a change coefficient of 1.35 over the Lez catchment for the 99.9[th] quantile of RCP8.5 represents an increase from 140 mm day[-1] to 189 mm day[-1].

## 6 Discussion

The present study assessed the intensification of extreme precipitation events under climate change on small Mediterranean catchments using high resolution RCMs simulations (~12 km) from the EMCORDEX exercise. It was shown that over the past period (1976-2005), EVAL simulations (RCM driven by ERAI) and HIST simulations (RCM is driven by a GCM) underestimate extreme events with respect to SAFRAN dataset.



This underestimation is potentially more severe as SAFRAN is itself deemed to underestimate observed extreme precipitation events (Quintana-Seguí et al., 2008a). Indeed, complementary analysis (not shown here) on the Aude and Lez catchments highlighted that SAFRAN underestimates precipitation quantiles beyond 95[th] with respect to pluviometer data from Météo-France network. It should be noted that the comparison between the SAFRAN gridded product and the

pluviometer sparse product is challenging and that these products are not independent. Yet, a perspective for the present study stands in a more representative computation of observed quantiles from pluviometer data set in place of SAFRAN ones. This paves the way towards a time varying quantile classification with an intra-daily cycle, for instance using 3-hour pluviometers data. Such intra-daily cycle could also be represented in the past/future change coefficients using 3-hour EMCORDEX precipitation outputs. Working on higher temporal resolution data (observation and simulation) would lead to

a futurisation method more consistent with flash flood time scales than the daily current method.

Future changes in precipitation are assessed through a multi model approach focusing on the mean and standard deviations of the EMCORDEX ensemble. Advanced ensemble statistics could be used in place of metrics based on the arithmetic mean and standard deviation, for instance with clustering methods based on model performances (Monerie et al., 2016). Though, as discussed in Reifen and Toumi (2009) and Knutti et al. (2010), model performances in the past do not necessarily relate

with model performances in the future and the choice of the criteria for multi-model studies should be further investigated.

As previously explained in Fig. 6, the monthly averaged precipitation simulated by the RCP8.5 ensemble decreases with respect to the HIST monthly averaged precipitation over the past period, especially from April to October for Lez, Aude and Muga catchments. For Aude and Muga boxes, there is a shift in the annual cycle; the peak occurs earlier in spring (in April instead of May). This suggests that a change in precipitation amplitude and in temporal seasonality should be expected.

Llasat and Puigcerver (1997) explained that intense precipitation events are mostly due to convective rainfall in autumn and to global circulation in spring. The maximum number of convective days and ratio between convective and total precipitation is recorded between August and September in la Boadella, and a positive trend (5% error level) in the annual number of convective days has been founded in the Muga catchment (Llasat et al, 2016). The change in spring precipitation is thus most likely due to changes in global fluxes that have different impacts on western coastal regions (Aude and Muga)

and on southern coastal regions (Cevennes). Indeed, Nuissier et al. (2011) explained that strong precipitation events over the west coast of the Mediterranean are mostly correlated with easterly fluxes, while strong precipitation events in the northern coastal region are correlated with south to southeasterly fluxes. Cassou et al. (2016) reported that climate changes will have a greater impact on easterly fluxes; this could explain the change in the annual cycle observed in Aude and Muga regions only. This hypothesis should be validated with an analysis of changes in geopotential fields.

**Conclusion**

A "futurization" approach is presented in this article; it consists in the computation of a past/future change coefficients applied to quantile-ranked RCM precipitation outputs. We use the EMCORDEX ensemble to estimate the model and





scenario uncertainty. The study focuses over the Lez, Aude and Muga Mediterranean catchments.  As a first step, EMCORDEX models' skills are evaluated in terms of mean and extreme precipitation over the present climate period 1976-2005. This analysis legitimates the use of the EMCORDEX models for past/future changes assessment.

It has been shown that cumulative precipitation is overestimated over the mountainous regions and underestimated over the coastal regions in autumn. Extreme events beyond 95[th] quantiles are underestimated. GCM forcing tends to enhance RCMs' underestimation of extreme precipitation events over the three catchments. Climate change impact is investigated from the RCP4.5 and RCP8.5 EMCORDEX simulations. In comparison with the present period, the monthly averaged precipitation decreases in spring and summer for RCP8.5. Past/future change coefficients computed from the EMCORDEX ensemble display an increase in extreme events precipitation intensity (beyond 90[th] quantile). This result is stronger over the end of 21[th] century, for RCP8.5, for all catchments of interest: all models within the ensemble agree on change coefficients larger than 1. The multi model approach developed here allows quantifying the uncertainty related to the past/future change coefficients. As a major conclusion of this study, we have shown with a high degree of confidence that all RCM models in EMCORDEX ensemble forecast an increase in extreme precipitation events.

As a perspective, change coefficients will be used to provide a kind of "futurize" extreme precipitation events that occurred in the past. In further studies, the hydrological impact of these "futurized" precipitations will be assessed using rainfall-runoff models over the Lez, Aude and Muga catchments. This generic method could also be applied to other catchments.

**Acknowledgements**

This work is an initiative of WSP-France. The financial support of this work has been provided by WSP-France under the CIFRE contract 2015/005. This paper was written under the framework of the International HYMEX project and the Spanish HOPE (CGL2014-52571-R) project. We acknowledge the World Climate Research Programme's Working Group on Regional Climate, and the Working Group on Coupled Modelling, former coordinating body of CORDEX and responsible panel for CMIP5. We also thank the climate modelling groups (listed in Table 1 of this paper) for producing and making available their model output. We also acknowledge the Earth System Grid Federation infrastructure an international effort led by the U.S. Department of Energy's Program for Climate Model Diagnosis and Intercomparison, the European Network for Earth System Modelling and other partners in the Global Organisation for Earth System Science Portals (GO-ESSP). The authors wish to acknowledge Samuel Somot (CNRM) for his fruitful discussions about regional modeling and the MedCORDEX database. Special thanks are also given to Jean-Marc Lacave and Gwenaëlle Hello from Météo-France for providing the daily and 3-hours pluviometers data, and to Céline Vargel for her useful work on the comparison between SAFRAN and pluviometers data.



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





# Tables

| | RCM | ALADIN5.2 | ALADIN5.3 | RCA4 | | | HIRHAM5 | RAMO22E |
|---|---|---|---|---|---|---|---|---|
| | Institution | Météo-France | Météo-France | SMHI | | | DMI | KNMI |
| | **Boundary layer scheme** | Ricard and Royer 1993 | Ricard and Royer 1993 | Cuxart et al. 2000 | | | Louis 1979 | Lenderink and Holtslag 2004; Siebesma |
| | **Convection** | Mass flux Bougeault 1985 | Mass flux Bougeault 1985 | Kain and Fritsch 1990, 1993; Kain 2004; Jones and Sanchez 2002 | | | Tiedtke 1989 | Tiedtke 1989; Nordeng 1994; Neggers et al. 2009 |
| | **Main references** | Colin et al. 2010; Hermann et al. 2011 | Colin et al. 2010; Hermann et al. 2011 | Samuelsson et al. 2011; Kupiainen et al. 2011 | | | Christensen et al. 2008 | Meijgaard van et al. 2012 |
| **Lateral Boundary Conditions** | **Evaluation simulation** | ERA-INTERIM (Reanalysis) | | | | | | |
| | **Historical simulation** | CNRM-CERFACS-CNRM-CM5 | | | MOHC-HadGEM2-ES | MPI-M-MPI-ESM-LR | ICHEC-EC-EARTH | |
| | **RCP4.5 simulation** | | | | | | | |
| | **RCP8.5 simulation** | | | | | | | |
| | **GCM members** | r8i1p1 | r1i1p1 | r1i1p1 | r1i1p1 | r1i1p1 | r12i1p1 | r3i1p1 | r1i1p1 |
| | **Name of pair GCM_RCM** | **CNRM-CM5 _ALADIN52** | **CNRM-CM5 _ALADIN53** | **CNRM-CM5 _RCA4** | **MOHC_ RCA4** | **MPI_ RCA4** | **ICHEC_ RCA4** | **ICHEC_ HIRHAM5** | **ICHEC_ RACMO22E** |

**Table 1: The parameterization and physics scheme related to the RCM (0.11°) used in the present study are presented in the five first lines. The lateral boundary conditions (LBC) for each type of simulations are presented in the boundary conditions with four different GCMs of CMIP5. The last row lists the name used for each GCM-RCMs pair throughout this study.**



# Figures

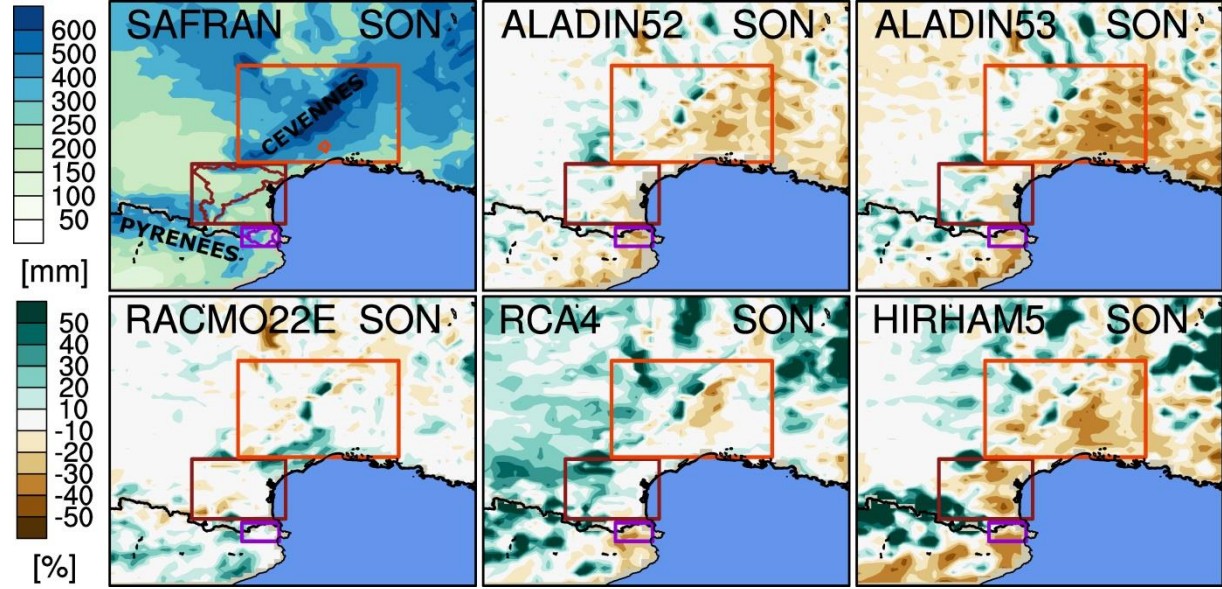

**Figure 1: Mean relative seasonal precipitation bias (%) in all the RCMs forced with ERA-Interim for the period 1981-2010. The top left panel shows the horizontal pattern of seasonal precipitation provided by the SAFRAN reference dataset (mm season⁻¹). Only the September-October-November (SON) season is shown here. The colored rectangles represent the regional boxes. The Cevennes box and the Lez catchment are in red; the Aude box and catchment in brown; and the Muga box and catchment in purple.**





**Figure 2: a-Annual cycle of precipitation over Aude box with SAFRAN dataset b-Bias of the annual cycle of precipitation simulated by the 5 RCMs with respect to SAFRAN's annual cycle of precipitation for the period 1981-2010 (no units), over the Aude box. The colored lines traduce the biases of RCMs simulated precipitation with respect to SAFRAN dataset (black line).**





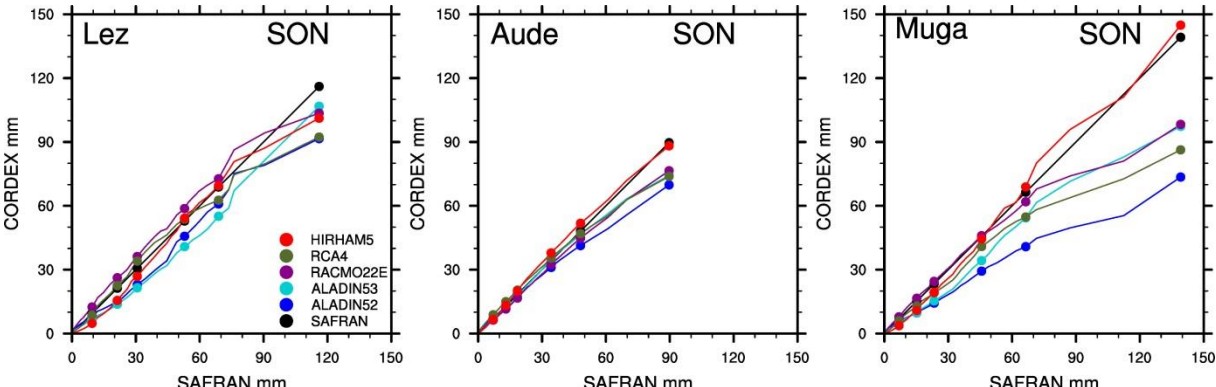

**Figure 3: Quantile-quantile diagram of daily precipitation in the cells of the 3 catchments for the period 1981-2010. EVAL simulated precipitations (colored lines) are compared to SAFRAN (black line). The x axis represents the precipitation quantile values with respect to the SAFRAN reference data. The y axis represents the precipitation intensity simulated by the EMCORDEX models for the same quantiles. If a colored line is above/below the black line, the corresponding RCM over/under-estimates quantile intensities with respect to SAFRAN. Units are in mm day[-1]. The colored dots represent, from left to right, the 90[th], 95[th], 97[th], 99[th], 99.5[th], 99.9[th] quantiles.**





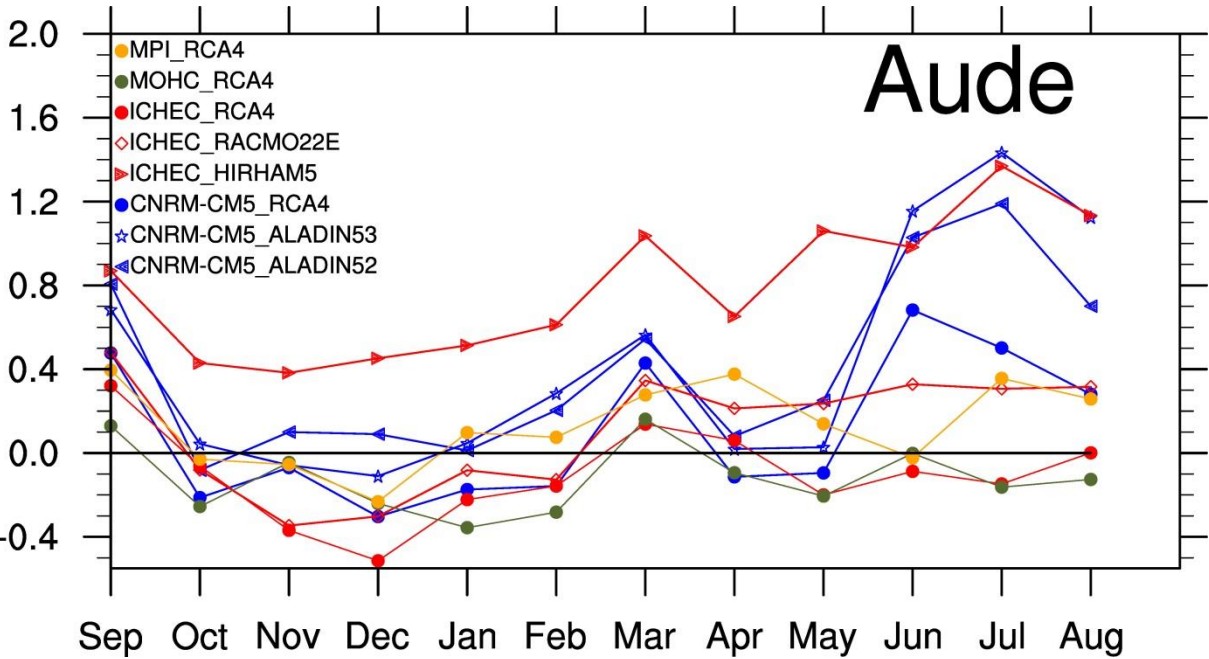

**Figure 4: Bias on the annual cycle RCMs simulated precipitation induced by GCMs lateral boundary conditions. Aude regional box is shown here for the period 1976-2005. This bias is estimated through the computation of the difference between HIST and EVAL simulated precipitation with respect to SAFRAN. ΔB (no units) represent this bias with: $\Delta B = \frac{HIST - SAFRAN}{SAFRAN} - \frac{EVAL - SAFRAN}{SAFRAN}$.**

5  **Colored lines refer to the GCMs and the markers refer to the RCMs drove. If a colored line is above/below the black line, the corresponding GCM induce an over/under-estimation of the precipitation simulated by the RCM.**

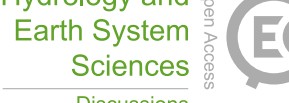



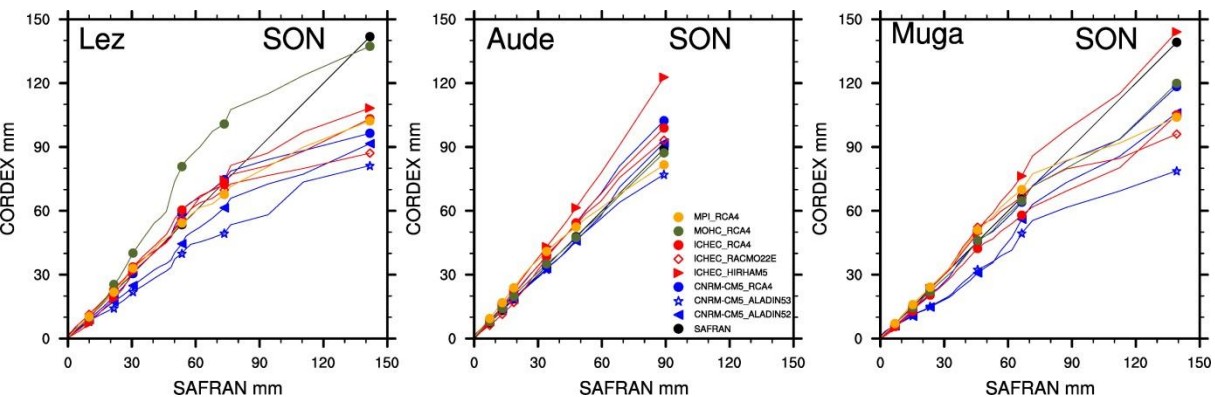

**Figure 5: Quantile-quantile diagram of daily precipitation in the cells of the 3 catchments for the period 1981-2010. HIST simulated precipitations (colored lines) are compared to SAFRAN (black line). The x axis represents the precipitation quantile values with respect to the SAFRAN reference data. The y axis represents the precipitation intensity simulated by the EMCORDEX**

5 **models for the same quantiles. If a colored line is above/below the black line, the corresponding RCM over/under-estimates quantile intensities with respect to SAFRAN. Units are in mm day[-1]. The colored dots represent, from left to right, the 90[th], 95[th], 97[th], 99[th], 99.5[th], 99.9[th] quantiles.**





**Figure 6 :** Annual cycle of mean precipitation (mm month$^{-1}$) simulated in the three regional boxes (Cevennes, Aude, Muga see figure 1). Solid lines represent the ensemble means computed amongst the eight GCM-RCM pairs. The shaded areas represent the ensemble spreads characterized by the standard deviation. HIST ensemble mean (1976-2005) is plotted in blue. RCP4.5 and RCP8.5 ensemble means (2071-2100) are respectively plotted in green and red. SAFRAN annual cycle computed over 1981-2010 is plotted in black.





**Figure 7: Change coefficients (Aq$_i$) over the [90-99.9] quantile range computed for each GCM-RCM pairs over the Lez, Aude and Muga catchments. The no change line (ai=1) is also displayed in solid black. The thick solid lines represent the ensemble mean for Aq$_i$, and the associated ensemble spread is represented by the shaded areas (standard deviation). RCP4.5 and RCP8.5 are respectively plotted in blue and red. For clarity purpose the scale of the x-axis was distorted according to the quantile discretization.**