# Peer review of "Evaluation of uncertainties in mean and extreme precipitation under climate changes for northwestern Mediterranean watersheds from high-resolution Med and Euro-CORDEX ensembles"

_Hydrology and Earth System Sciences, 2017_

## Referee Comment (RC1) · Anonymous Referee #1 · 6 Jun 2017

The manuscript addresses the issue of changes in precipitation patterns under climate change in three selected Mediterranean regions, using a CORDEX high-resolution ensemble. The topic is dealt using widely accepted methodologies (evaluation metrics) and some newer concepts for quantifying changing of extreme precipitation patterns and error additivities in GCM/RCM simulations. The paper in general is well written and constructed. The abstract and conclusions summarize the basic features and findings of the work presented. Their introduction, despite being a bit lengthy is quite

informative, the methodology clearly presented (some issues addressed below) and the description of results clear and concise.

My major comment is that the ensemble members used in this study do not cover the existing EURO and MED-cordex simulations, as the title of the manuscript indicates. The criteria for not including existing and most importantly independent EURO/MED CORDEX simulations (eg RegCM4 or WRF331F) is not clear to me. Moreover, the authors decided to include 2 ensemble members from the same family (ALADIN5.2 and ALADIN5.3) i.e. two model versions which I expect they share similar structural errors and therefore expected to share similar behaviour. I don't find this choice methodologically sound. I understand the choice of authors, only if additional independent EURO-MED CORDEX ensemble members were not available by the time of manuscript preparation.

Technical corrections Page 1, Line 18: "over past period" Over the past period: which is this period?

Page 4, Line 14: there is a submitted paper, if available please provide the full citation

Page 5, Line 7. I missed two important ensemble members of EURO/MED CORDEX simulations, namely RegCM, and WRF. Especially RegCM is one of the most traditional regional climate models used for the investigation of European and particularly Mediterranean climate and I was wondering why authors did not include those ensemble members in their current study.

Page 6, line 4: I don't understand why the RCMs with spatial resolution of 12 Km where regridded to the 8 Km of SAFRAN. Why didn't they regrid from 8 to 12 Km?

Page 6, line 5. Remapping procedures are known to affect precipitation statistics (e.g. Diaconescu et al., 2015 http://journals.ametsoc.org/doi/pdf/10.1175/JHM-D-15-0025.1). The authors mention that they have tested how interpolation methods affect their results, without providing additional information. Extra care needs to be taken,

especially when one attempts a percentile analysis in precipitation.

Page 6, line 25. Could authors add a couple of lines on the behaviour of $\Delta\hat{I}\check{S}$? What it means when $\Delta\hat{I}\check{S}$ is <0 or >0? Is shortly mentioned in Figure 4 caption, better mention in text.

Page 8, line 9." Figure 2b displays the normalized annual cycle...". The caption of Figure 2b says "Bias of the annual cycle of precipitation".

Page 9, line 7."The results are coherent with other studies...". Please refer to those other studies. Page 9, line 13: "thus" > eventually mean "those"?

Page 20, Table 1: I miss the Radiation, Microphysics and Land Surface Model selections of each RCM simulation. It is useful information for regional climate modellers.

Page 9, line 20: Deque et al., 2011 is missing in the references list.

Page 10, line 9. "...Fig 5 is considerably larger". I don't find the differences in spreads between Fig 3 and 5 "considerable larger" for the Muga region.

Page 11, line 3-4: "Future precipitations from RCP... distribution". I don't think I understand this sentence.

Page 12 ,line 14-15. While some reported that model performance in the past do not necessarily relate with model performance in the future, some report the opposite: Boberg and Christensen, 2012, Nature Climate Change.

Comparison of Fig 3 and Fig 5 is a bit confusing. In Figure 5, colors are used for GCMs and markers for RCMs, which is quite nice. In Figure 3, colors are used for RCMS; it would be easier to keep using markers for RCMs, similar to Figure 5. Finally, is there a particular need to use SAFRAN in Figure 3 and 5? Isn't it supposed to be the diagonal line?

Figure 7. If this figure refers to autumn it should be mentioned in the figure caption.

[Figure]

---

## Short Comment (SC1) · 22 Jun 2017

J.-H. Yoon

yjinho@gist.ac.kr

Major comments 1. It is well written manuscript with throughly analysis. 2. On the other hand, I'd like to question the validity of the simulated changes given the fact the all RCMs underestimate extreme precipitation even in EVAL. Further, this underestimation seems worse in HIST. Can the authors provide quantitative assessment how different in EVAL and HIST? 3. Continued... Can the authors separate one with better/worse performance in terms of extreme precipitation? 4. Uncertainty in observation - SAFRAN: In discussion section, the authors provide a bit of confusing message about how good observation dataset is. If there are different high-quality observation datasets, it'd be nice to provide them.

Overall, it is well written and organized paper. However, I hope the questions/comments will be taken care of before its publication.

---

## Author Comment (AC1) · 24 Jul 2017

Dear Editor,

We are responding to the comments from Reviewer 1 who has provided a very long list of interesting comments.

The major question about the model ensemble has been justified and the other technical corrections have been answered.

Thereafter, the comments from Reviewer 1 start with "RC1:" and our responses start with "ACD:"

RC1: The manuscript addresses the issue of changes in precipitation patterns under climate change in three selected Mediterranean regions, using a CORDEX high-resolution ensemble. The topic is dealt using widely accepted methodologies (evaluation metrics) and some newer concepts for quantifying changing of extreme precipitation patterns and error additivities in GCM/RCM simulations. The paper in general is well written and constructed. The abstract and conclusions summarize the basic features and findings of the work presented. Their introduction, despite being a bit lengthy is quite informative, the methodology clearly presented (some issues addressed below) and the description of results clear and concise. My major comment is that the ensemble members used in this study do not cover the existing EURO and MED-cordex simulations, as the title of the manuscript indicates. The criteria for not including existing and most importantly independent EURO/MED CORDEX simulations (eg RegCM4 or WRF331F) is not clear to me. Moreover, the authors decided to include 2 ensemble members from the same family (ALADIN5.2 and ALADIN5.3) i.e. two model versions which I expect they share similar structural errors and therefore expected to share similar behaviour. I don't find this choice methodologically sound. I understand the choice of authors, only if additional independent EUROMED CORDEX ensemble members were not available by the time of manuscript preparation.

ACD: We are very grateful that Reviewer 1 has read our paper very carefully and has left comments very precisely. Indeed, the major question is frequently asked and the response is as follows. The EUROMED CORDEX ensemble establishment has been driven by several reasons. First of all, and as supposed by the reviewers, no more ensemble members could be downloaded after the ESGF website hacking in August 2015. The website crash hasn't been available for more or less one year. After this period, the paper was already being written, and considering the above reasons, we

have decided to maintain the same model ensemble. The final aim of this study is to focus on the hydrological impact of future precipitation issued from the quantile ranked change coefficient function. Thus, we are not really arguing to assess the entire climate model spread as studied in the CMIP5 ensemble by McSweeney et al. (2015). Finally, ALADIN5.2 (Med-CORDEX) and ALADIN5.3 (Euro-CORDEX) are both effective to the same research laboratory, e.g. the CNRM. However, as discussed with Samuel Somot (personal communication), despite they both are issue from the same framework, they have some different physical structures and different parameters • Binary changes (V6.01) because the calculator had changed. • RRTM for the LW • FMR-6 bands for the SW • ECUME for the air-sea fluxes • The mix length like for Lenderink ALADIN5.2 is more precisely described in Colin et al. (2010) and Herrman et al. (2011) and the ALADIN5.3 is slightly described in Tramblay et al. (2017) and Bador et al. (2017). Moreover, these two regional climate models are the most used in France, so it was interesting to propose their evaluation with an assessment of the added value with the most recent one (ALADIN5.3) in terms of mean and extreme precipitation over complex regions.

RC1: Technical corrections

RC1: Page 1, Line 18: "over past period" Over the past period: which is this period?

ACD: The period referred here is 1981-2010. We propose to add into the abstract.

RC1: Page 4, Line 14: there is a submitted paper, if available please provide the full citation

ACD: Unfortunately, the paper is not published at this date. It has been submitted in the Journal of Hydrology.

RC1: Page 5, Line 7. I missed two important ensemble members of EURO/MED CORDEX simulations, namely RegCM, and WRF. Especially RegCM is one of the most traditional regional climate models used for the investigation of European and particularly Mediterranean climate and I was wondering why authors did not include those ensemble members in their current study.

ACD: The absence of WRF and RegCM models are due to the same reasons explained in the answer of the major comments.

RC1: Page 6, line 4: I don't understand why the RCMs with spatial resolution of 12 Km where regridded to the 8 Km of SAFRAN. Why didn't they regrid from 8 to 12 Km?

ACD: The final aim of this work consists in applying future precipitation on hydrological models to assess their impacts as explained in section 1. Hydrological models generally work at 100m resolution. The spatial scale is different between the precipitation input resolution and the model computing resolution. It is a source of error that has to be minimizing as much as possible. So, to go closer to the hydrological scale, we decided to regrid the precipitation from 12 to 8km. To ensure that this operation will not bring additional biases to the precipitation an analytical test has been done.

RC1: Page 6, line 5. Remapping procedures are known to affect precipitation statistics (e.g. Diaconescu et al., 2015 http://journals.ametsoc.org/doi/pdf/10.1175/JHM-D-15-0025.1). The authors mention that they have tested how interpolation methods affect their results, without providing additional information. Extra care needs to be taken, especially when one attempts a percentile analysis in precipitation.

ACD: An analytical equation, containing latitude and longitude parameters, is resolved over the 8 km grid and the 12 km grid. Then the analytical field from the 12 km grid is interpolated to the 8 km grid. The previous analytic field on the 8km grid is then compared to the interpolated analytic field. The analytic equation is: f(x) = 2 - cos(pi*(acos(cos(lat*x)*cos(lon*x))/alt)) The attached figure show the function resolved on the top and the spatial interpolation biases on the bottom. The bottom map shows biases from 4.10-6 to 5.10-5 for the interpolation method used (Bilinear) over the whole domain studied. Then we have considered that the interpolation impact was negligible. We can provide additional information in section 2.3

RC1: Page 6, line 25. Could authors add a couple of lines on the behaviour of Ί? What it means when Ί is <0 or >0? Is shortly mentioned in Figure 4 caption, better mention in text.

ACD: The deltaB corresponds to the bias in the annual cycle of precipitation simulated with the RCMs that is strictly related to the influence of the lateral boundary condition imposed by the GCM. A high positive value indicates an overestimation of the total monthly precipitation, and a negative value indicates an underestimation of the total monthly precipitation. We propose to include this additional information in section 2.4.

RC1: Page 8, line 9." Figure 2b displays the normalized annual cycle". The caption of Figure 2b says "Bias of the annual cycle of precipitation".

ACD: We thank the review of this precise notification. We propose to correct this mistake by replacing the term "normalized" with "bias of the" like expressed in the caption.

RC1: Page 9, line 7." The results are coherent with other studies". Please refer to those other studies.

ACD: This sentence has been misunderstood. The role was to outline the following sentences where specific results are compared to other studies. We propose to replace this sentence by "The results are coherent with specific studies as cited thereafter".

RC1: Page 9, line 13: "thus" > eventually mean "those"?

ACD: We thank the reviewer to have noticed that mistake. We propose to correct it as recommended by the reviewer.

RC1: Page 20, Table 1: I miss the Radiation, Microphysics and Land Surface Model selections of each RCM simulation. It is useful information for regional climate modelers.

ACD: This information is not actually available on the website where the data have been downloaded and on the Med and Euro-CORDEX websites.

RC1: Page 9, line 20: Déqué et al., 2011 is missing in the references list.

ACD: We thank the reviewer for this comment is due to a mistake. The corresponding paper has to be cited as Déqué et al., 2012.

RC1: Page 10, line 9. "...Fig 5 is considerably larger". I don't find the differences in spreads between Fig 3 and 5 "considerable larger" for the Muga region.

ACD: We agreed that the adjective used isn't adapted to the reality observed on the plot. To correct it, we propose to replace this formulation with "...Fig5 is larger except for the Muga catchment".

RC1: Page 11, line 3-4: "Future precipitations from RCP...distribution". I don't think I understand this sentence.

ACD: Indeed, the idea that we would like to express here is not clearly stated through this sentence. To correct it, we propose to replace the sentence with: "Precipitation issued from RCP simulations are not bias corrected here. However, since explained in section 1, they are used to estimate to change coefficients between past and future quantile intensities of precipitation."

RC1: Page 12, line 14-15. While some reported that model performance in the past do not necessarily relate with model performance in the future, some report the opposite: Boberg and Christensen, 2012, Nature Climate Change.

ACD: This is an interesting article that we ignore. It will be relevant to add it to the discussion section.

RC1: Comparison of Fig 3 and Fig 5 is a bit confusing. In Figure 5, colors are used for GCMs and markers for RCMs, which is quite nice. In Figure 3, colors are used for RCMS; it would be easier to keep using markers for RCMs, similar to Figure 5.

ACD: We understand that similar markers would make easier the interpretation of the EVAL and HIST quantile-quantile plots. Unfortunately, that change cannot be done

easily during the summer. If the Editor considers that make this change is kind of major importance, we will try to do it as soon as possible, but it should not be done before September 2017.

RC1: Finally, is there a particular need to use SAFRAN in Figure 3 and 5? Isn't it supposed to be the diagonal line?

ACD: As supposed by the reviewer, the SAFRAN quantiles and the diagonal line are superposed in the quantile-quantile plots. However, it's interesting to see the specific dots values for the SAFRAN quantiles. We prefer to maintain that figure like that for this paper as we consider that the proposed modification won't have a significant change in the figure interpretation.

RC1: Figure 7. If this figure refers to autumn, it should be mentioned in the figure caption.

ACD: As recommended by the reviewers, we propose to insert in the figure caption that the change coefficient plots correspond to the autumn season (SON).

Yours sincerely,

Antoine Colmet-Daage

On behalf of all the co-authors.

Bibliography:

Bador, M., Terray, L., Boe, J., Somot, S., Alias, A., Gibelin, A.-L., Dubuisson, B., 2017. Future summer mega-heatwave and record-breaking temperatures in a warmer France climate. Environmental Research Letters.

Colin, J., Déqué, M., Radu, R., Somot, S., 2010. Sensitivity study of heavy precipitation in Limited Area Model climate simulations: influence of the size of the domain and the use of the spectral nudging technique. Tellus A 62, 591–604.

Déqué, M., Somot, S., Sanchez-Gomez, E., Goodess, C.M., Jacob, D., Lenderink, G., Christensen, O.B., 2012. The spread amongst ENSEMBLES regional scenarios: regional climate models, driving general circulation models and interannual variability. Climate Dynamics 38, 951–964. doi:10.1007/s00382-011-1053-x

Herrmann, M., Somot, S., Calmanti, S., Dubois, C., Sevault, F., 2011. Representation of spatial and temporal variability of daily wind speed and of intense wind events over the Mediterranean Sea using dynamical downscaling: impact of the regional climate model configuration. Natural Hazards and Earth System Sciences 11, 1983–2001.

McSweeney, C.F., Jones, R.G., Lee, R.W., Rowell, D.P., 2015. Selecting CMIP5 GCMs for downscaling over multiple regions. Climate Dynamics 44, 3237–3260.

Tramblay, Y., Hertig, E.: Modelling extreme dry spells in the Mediterranean region in connection with atmospheric circulation. Under review in Atmospheric Research
* * *
Interactive
comment

RCM_Regrid_Safran_ErreurInterpolation_bilinear_Analytique

[Figure]

**Fig. 1.** Error of spatial interpolation through the ESMF Bilinear method

---

## Author Comment (AC2) · 24 Jul 2017

Dear Editor,

We are responding to the comments from Reviewer 2 who has provided interesting comments.

The major comments have been answered with additional elements.

Thereafter, the comments from Reviewer 2 start with "RC2:" and our responses start with "ACD:"

RC2: Major comments

RC2: 1. It is well written manuscript with throughly analysis.

RC2: 2. On the other hand, I'd like to question the validity of the simulated changes given the fact the all RCMs underestimate extreme precipitation even in EVAL. Further, this underestimation seems worse in HIST. Can the authors provide quantitative assessment how different in EVAL and HIST?

ACD: We understand perfectly the reviewer doubts about the validity of the simulated changes given the RCM underestimation of extreme precipitation. However, we would like to specify that the future changes presented are relative, indeed it represents a ratio between past and future precipitation. Thus, the past underestimation must be effective in future simulations too, but the relative change between both won't be affected. Moreover, the papers from Monerie et al. (2016), Reifen and Toumi (2009) and Knutti et al. (2010) cited in section 6 shows that past performance does not guaranty future accuracy, and are not related to the relative changes. Then, a quantitative assessment between the extreme precipitation underestimation of EVAL and HIST simulation isn't realistic. Despite we could make the hypothesis of cumulative errors between the GCMs and the RCMs for the mean annual cycle of precipitation, we cannot make the same hypothesis for extreme precipitation because the physical processes of those phenomena are very different and not linear.

RC2: 3. Continued... Can the authors separate one with better/worse performance in terms of extreme precipitation?

ACD: Despite some systematic performances seems to appear through the metrics describing models performances to simulate mean and extreme precipitation, the aim of this paper isn't to classify models among themselves but rather assess their uncertainties. Moreover, it would not be robust to discriminate models according to their performances in such specific regions with these specific metrics chosen according to the particular goal of this study.

RC2: 4. Uncertainty in observation - SAFRAN: In discussion section, the authors provide a bit of confusing message about how good observation dataset is. If there are different high-quality observation datasets, it'd be nice to provide them.

ACD: The message in the discussion section attempts to alert on the SAFRAN underestimation of extreme precipitation compared to the pluviometers data from the studied catchments and according to Quintana-Seguí et al. (2008). The attached figure shows this SAFRAN seasonal underestimation for the Lez catchment comparing the data from the 3 grid cells covering the catchment (black line) and the pluviometers available in the catchment (colored lines). However, as explained in the section 2.3, SAFRAN is the better gridded observation dataset covering the studied regions. Moreover, it is a reanalysis that have been done precisely to force the hydrological and soil models later.

RC2: Overall, it is well written and organized paper. However, I hope the questions/comments will be taken care of before its publication.

Yours sincerely,

Antoine Colmet-Daage

On behalf of all the co-authors.

Bibliography:

Knutti, R., Furrer, R., Tebaldi, C., Cermak, J., Meehl, G.A., 2010. Challenges in combining projections from multiple climate models. Journal of Climate 23, 2739–2758. doi:10.1175/2009JCLI3361.1

Monerie, P.-A., Sanchez-Gomez, E., Boé, J., 2016. On the range of future Sahel

precipitation projections and the selection of a sub-sample of CMIP5 models for impact studies. Climate Dynamics 1–20. doi:10.1007/s00382-016-3236-y

Quintana-Seguí, P., Le Moigne, P., Durand, Y., Martin, E., Habets, F., Baillon, M., Canellas, C., Franchisteguy, L., Morel, S., 2008. Analysis of Near-Surface Atmospheric Variables: Validation of the SAFRAN Analysis over France. J. Appl. Meteor. Climatol. 47, 92–107. doi:10.1175/2007JAMC1636.1

Reifen, C., Toumi, R., 2009. Climate projections: Past performance no guarantee of future skill? Geophysical Research Letters 36. doi:10.1029/2009GL038082

[Figure]

**Fig. 1.** Seasonal extreme precipitation quantiles underestimation of SAFRAN dataset compared to local pluviometers.

---

## Author Response (AR1)

Dear Editor,

We appreciate the extensive and relevant comments made by Reviewer 1 and Reviewer 2. Thereafter, RC1/RC'2 s questions and remarks are numbered and cited in italics (paragraphs denoted by RC1/RC2) and our responses are denoted by ACD. Complementary illustrations are provided when necessary. Changes in the article were made accordingly in revision mode.

**Response to RC1**
**Major comments:**
**Q1 (RC1) - Ensemble members**
*The manuscript addresses the issue of changes in precipitation patterns under climate change in three selected Mediterranean regions, using a CORDEX high-resolution ensemble.*
*The topic is dealt using widely accepted methodologies (evaluation metrics) and some newer concepts for quantifying changing of extreme precipitation patterns and error additivities in GCM/RCM simulations. The paper in general is well written and constructed. The abstract and conclusions summarize the basic features and findings of the work presented. Their introduction, despite being a bit lengthy is quite informative, the methodology clearly presented (some issues addressed below) and the description of results clear and concise.*
*My major comment is that the ensemble members used in this study do not cover the existing EURO and MED-cordex simulations, as the title of the manuscript indicates. The criteria for not including existing and most importantly independent EURO/MED CORDEX simulations (eg RegCM4 or WRF331F) is not clear to me. Moreover, the authors decided to include 2 ensemble members from the same family (ALADIN5.2 and ALADIN5.3) i.e. two model versions which I expect they share similar structural errors and therefore expected to share similar behaviour. I don't find this choice methodologically sound. I understand the choice of authors, only if additional independent EUROMED CORDEX ensemble members were not available by the time of manuscript preparation.*

*Page 5, Line 7. I missed two important ensemble members of EURO/MED CORDEX simulations, namely RegCM, and WRF. Especially RegCM is one of the most traditional regional climate models used for the investigation of European and particularly Mediterranean climate and I was wondering why authors did not include those ensemble members in their current study.*

**ACD:**
RC1 recommended that additional members were added to the study. This is a crucial point that was also highlighted by the editor. Though the authors would ideally enlarge the ensemble, this is not feasible in the context of this study.

Indeed, the ensemble list was established at the beginning of A. Colmet-Daage's (ACD) PhD work in May 2015. At that time, only a few members were available for the EUROMED-CORDEX exercise on the ESGF server. The members presented in this study were downloaded before August 2015 when the web site was hacked and, as a consequence, down for about a year. The study was then carried out with a limited list and the paper written accordingly. As of today, the ESGF website is back in service and now hosts additional members. Amongst these new members, only three (REMO2015, CCLM4-8-17 and WRF311F) meet with the criteria requested for our work :
- Spatial resolution of 0.11 degree,
- Availability of the two emissions scenarios RCP4.5 and RCP8.5
- Availability of the simulations in the future, from 2011 to 2100.

Nevertheless, we have checked that our current ensemble displays a significant dispersion, with different members. Figure 1 displays the quantile change coefficient calculated for each model separately (no change line in black and GCM-RCM pairs in colored lines). RCMs are distinguished by the line color while the GCMs are distinguished by the line style. Each member is different from the other member, with a homogeneous spread. We thus consider that adding 3 members to the ensemble is not a crucial point and does not discredit our analysis.

[Figure]

**Figure 1: Seasonal change coefficients (Aqi) over the [95-99.9] quantile range computed for each GCM-RCM pairs over the Lez catchment in 2071-2100 according to the RCP4.5 emission scenario. The no change line (ai=1) is displayed in solid black. The colored lines represent each RCM, and the different linetypes display the different GCMs.**

Differences between ALADIN5.2 and ALADIN5.3 are quite clear and even though these members share the same RCM name, they can still be considered as independent for our areas of interest. Despite they are derived from the same original source code, they differ in terms of parameters settings and physical scheme (S. Somot, personal communication):

- Binary changes (V6.01) because the calculator had changed.
- RRTM for the LW

- FMR-6 bands for the SW
- ECUME for the air-sea fluxes

For more details, ALADIN5.2 is more precisely described in Colin et al. (2010) and Herrman et al. (2011). ALADIN5.3 is briefly described in Tramblay et al. (2017) and Bador et al. (2017). It should be noted that these two regional climate models are the most commonly used in France, so it was interesting to propose their evaluation with an assessment of the added value of ALADIN5.3 in terms of mean and extreme precipitation.

To conclude, though we agree that including theses members would strengthen the study as in the CMIP5 ensemble by McSweeney et al. (2015), we estimate that downloading and analyzing the data would take several months. Since the beginning of this work some advances in the matter have been proposed in the bibliography. Then, we consider your recommendation as a good proposal for a future paper, enlarging the region of study and updating the methodology.

**Q2 (RC1) - Interpolation**
*Page 6, line 4: I don't understand why the RCMs with spatial resolution of 12 Km where regridded to the 8 Km of SAFRAN. Why didn't they regrid from 8 to 12 Km.*
*Page 6, line 5. Remapping procedures are known to affect precipitation statistics (e.g. Diaconescu et al., 2015 http://journals.ametsoc.org/doi/pdf/10.1175/JHM-D-15-0025.1). The authors mention that they have tested how interpolation methods affect their results, without providing additional information. Extra care needs to be taken, especially when one attempts a percentile analysis in precipitation.*

**ACD:**
The answer to this question stands in two points. Firstly, the spatial resolution of the multi-model ensemble is $0.11^\circ$, but RCMs operate on different grids. In order to compare the simulations to the SAFRAN dataset considered here as the reference, the precipitation fields must be projected onto a common grid, otherwise the differences respect to SAFRAN could not be computed. For this reason the precipitation field from the different RCMs was interpolated to the SAFRAN grid, by using the same interpolation technique.
Secondly, this research study aims at applying future precipitation on hydrological models to assess their impacts as explained in section 1. The resolution for hydrological models on our region of interest is about 100m, which is significantly smaller than the atmospheric model resolution. Regrinding the precipitation to the smaller scale (8km rather than 12 km) thus allows bringing the precipitation field closer to the hydrological scale.

It is important to quantify the errors due to interpolation. Hence, the impact of the interpolation step was investigated on an analytical function. The analytical function of latitude and longitude (f(x) = 2 - cos(pi*(acos(cos(lat*x)*cos(lon*x))/alt)), displayed in Fig. 2 – top panel) is solved over both the 8 km and 12 km grids. The 12-km field is interpolated onto the 8 km grid and compared with the 8-km field. The difference through the computation of the infinity norm between these 2 fields is displayed in Fig. 2 – bottom panel. It ranges between $4.10^{-6}$ to $5.10^{-5}$ when using a bi-linear interpolation. This error is considered as negligible for our application. Additional information was added in section 2.3

[Figure]

**Figure 2 : Error of spatial interpolation of an analytical function through the ESMF Bilinear method.**

**Q3 (RC2) – Underestimation of extreme precipitation**
*On the other hand, I'd like to question the validity of the simulated changes given the fact the all RCMs underestimate extreme precipitation even in EVAL. Further, this underestimation seems worse in HIST. Can the authors provide quantitative assessment how different in EVAL and HIST?*
*Continued... Can the authors separate one with better/worse performance in terms of extreme precipitation?*
We understand perfectly the reviewer doubts about the validity of the simulated changes given the RCM underestimation of extreme precipitation. However, we would like to specify that the future changes presented are relative, indeed it represents a ratio between past and future precipitation (HIST and RCP). Thus, even if extreme precipitations are underestimated in the past, the relative change between past and future remains more reliable than absolute ones. Papers from Reifen and Toumi (2009) and Knutti et al. (2010) cited in section 6 acknowledge that past performance does not guaranty future accuracy, and are not related to the relative changes. Whether the model performance in present climate affects its response to global warming is still an open question in the modeling community.

In this paper, we show the impact of the GCM lateral boundary conditions on the RCMs simulation of the mean annual cycle of precipitation. For that, the hypothesis of additive of GCMs and RCMs biases has been assumed, but only for the temporal means of the precipitation over the considered period, for which the large scale influence (GCM) can be much more noticeable. Please, note that the same additive assumption would be less applicable to extreme values of precipitation, since the occurrence of the precipitation extreme mainly results from non-linear effects.

Mean and extreme precipitation were assessed in this study over specific regions. A global evaluation of performance for precipitation amongst the models is beyond the scope of this study.

**Q4 (RC2) – Using SAFRAN as a reference**
*Uncertainty in observation - SAFRAN: In discussion section, the authors provide a bit of confusing message about how good observation dataset is. If there are different high-quality observation datasets, it'd be nice to provide them.*

**ACD**
The SAFRAN is the best gridded observation dataset available to the community, covering the studied region as explained in the section 2.3. Moreover, it was designed to force hydrological and soil models. It was used as a reference data set in several regional studies (Dayon, 2015 ; Vidal et al., 2010 ; Vrac et Friederichs, 2015).
Of course, SAFRAN is a reanalysis product and presents intrinsic biases. The message in the discussion section attempts to alert, as Quintana-Seguí et al. (2008) did before, on the SAFRAN underestimation of extreme precipitation. Complementary analysis was carried out to assess the reliability of SAFRAN on our region on interest as presented in Fig. 3. The seasonal quantile-quantile plots between SAFRAN extreme precipitation and the 4 local pluviometers are presented. The black line and dot symbols represent the SAFRAN precipitation quantiles for the 3 grid cells covering the Lez catchment, the colored lines with a triangular markers represent each pluviometer, and finally the green line with a rectangular marker represent the quantiles average over these 4 pluviometers. The underestimation of extreme precipitation in SAFRAN varies with the season, it can reach up to 30% for higher quantiles. Recently, Quintana-Seguí et al. (2017) have compared the daily precipitation produced by SAFRAN in Spain to another well-known product, Spain02 (which is not available in France), and to the observations. They show that the extreme daily precipitation produced by SAFRAN is not as good as Spain02's but the differences are small.

[Figure]

Figure 3: Seasonal extreme precipitation quantiles underestimation of SAFRAN dataset compared to local pluviometers.

**Technical corrections**
*RC1: Page 1, Line 18: "over past period" Over the past period: which is this period?*
ACD: The period (1981-2010) was added in the abstract.

*RC1: Page 4, Line 14: there is a submitted paper, if available please provide the full citation*
ACD: Unfortunately, the paper is not published at this date. It has been submitted in the Journal of Hydrology.

*RC1: Page 6, line 25. Could authors add a couple of lines on the behaviour of δŠ? What it means when δŠ is <0 or >0? Is shortly mentioned in Figure 4 caption, better mention in text.*
ACD: The deltaB corresponds to the bias in the annual cycle of precipitation simulated with the RCMs that is strictly related to the influence of the lateral boundary condition imposed by the GCM. A high positive value indicates an overestimation of the total monthly precipitation, and a negative value indicates an underestimation of the total monthly precipitation. This explanation was added in section 2.4.

*RC1: Page 8, line 9." Figure 2b displays the normalized annual cycle". The caption of Figure 2b says "Bias of the annual cycle of precipitation".*
ACD: The caption is correct. The mistake was corrected in the text.

*RC1: Page 9, line 7." The results are coherent with other studies". Please refer to those other studies.*
ACD: This sentence rephrased to "The results are coherent with specific studies as cited thereafter".

*RC1: Page 9, line 13: "thus" > eventually mean "those"?*

ACD: This was corrected in the text.

*RC1: Page 20, Table 1: I miss the Radiation, Microphysics and Land Surface Model selections of each RCM simulation. It is useful information for regional climate modelers.*
ACD: This information is neither currently available on the website where the data were downloaded nor on the Med and Euro-CORDEX websites.

*RC1: Page 9, line 20: Déqué et al., 2011 is missing in the references list.*
ACD: The reference was corrected to Déqué et al., **2012**.

*RC1: Page 10, line 9. "...Fig 5 is considerably larger". I don't find the differences in spreads between Fig 3 and 5 "considerable larger" for the Muga region.*
ACD: The sentence was rephrased to "…Fig5 is larger except for the Muga catchment".

*RC1: Page 11, line 3-4: "Future precipitations from RCP…distribution". I don't think I understand this sentence.*
ACD: The sentence was rephrased to :
"Precipitation issued from RCP simulations are not bias corrected here. However, since explained in section 1, they are used to estimate to change coefficients between past and future quantile intensities of precipitation."

*RC1: Page 12, line 14-15. While some reported that model performance in the past do not necessarily relate with model performance in the future, some report the opposite: Boberg and Christensen, 2012, Nature Climate Change.*
ACD: This relevant paper was cited in the discussion.

*RC1: Comparison of Fig 3 and Fig 5 is a bit confusing. In Figure 5, colors are used for GCMs and markers for RCMs, which is quite nice. In Figure 3, colors are used for RCMS; it would be easier to keep using markers for RCMs, similar to Figure 5.*
ACD: We understand that similar markers would make easier the interpretation of the EVAL and HIST quantile-quantile plots. Unfortunately, the comprehension of the figure looks much more difficult if we make these changes putting all the lines in black. The figure 3 joined shows the figure 3 with these changes. For the Aude catchment, gap between RCMs quantiles are significantly more difficult to distinguish. Thus, we propose to change the markers and conserve the line colors. The figure 4 shows this new version of the figure, and looks more easily comprehensible. We inserted this one into the revised manuscript.
Finally, to stay coherent with the figures comparing control simulations against historical simulations, we applied the same changes to the figure 2b showing the bias of the annual cycle of precipitation. The figure 5 shows this modification.

[Figure]

**Figure 4: Black line version of the figure 3 from the manuscript with the markers changed as expected by the review.**

[Figure]

**Figure 5 : Proposed correction of the figure 3 from the manuscript, with the markers changed but conserving the colored lines.**

[Figure]

**Figure 6 : Proposed correction of the figure 2 from the revised manuscript, with the markers changed but conserving the colored lines.**

*RC1: Finally, is there a particular need to use SAFRAN in Figure 3 and 5? Isn't it supposed to be the diagonal line?*
ACD: As suggested by the reviewer, the SAFRAN quantiles and the diagonal line are the same. The quantiles value for the SAFRAN are indicated for information purpose

*RC1: Figure 7. If this figure refers to autumn, it should be mentioned in the figure caption.*
ACD: As recommended by the reviewers, it was specified in the figure caption that it corresponds to the autumn season (SON).

Yours sincerely,

Antoine Colmet-Daage,
On behalf of all the co-authors.
* * *
Bibliography (RC1) :
Bador, M., Terray, L., Boe, J., Somot, S., Alias, A., Gibelin, A.-L., Dubuisson, B., 2017. Future summer mega-heatwave and record-breaking temperatures in a warmer France climate. Environmental Research Letters.

Colin, J., Déqué, M., Radu, R., Somot, S., 2010. Sensitivity study of heavy precipitation in Limited Area Model climate simulations: influence of the size of the domain and the use of the spectral nudging technique. Tellus A 62, 591–604.

Déqué, M., Somot, S., Sanchez-Gomez, E., Goodess, C.M., Jacob, D., Lenderink, G., Christensen, O.B., 2012. The spread amongst ENSEMBLES regional scenarios: regional climate models, driving general circulation models and interannual variability. Climate Dynamics 38, 951–964. doi:10.1007/s00382-011-1053-x

Herrmann, M., Somot, S., Calmanti, S., Dubois, C., Sevault, F., 2011. Representation of spatial and temporal variability of daily wind speed and of intense wind events over the Mediterranean Sea using dynamical downscaling: impact of the regional climate model configuration. Natural Hazards and Earth System Sciences 11, 1983–2001.

McSweeney, C.F., Jones, R.G., Lee, R.W., Rowell, D.P., 2015. Selecting CMIP5 GCMs for downscaling over multiple regions. Climate Dynamics 44, 3237–3260.

Tramblay, Y., Hertig, E.: Modelling extreme dry spells in the Mediterranean region in connection with atmospheric circulation. Under review in Atmospheric Research

Bibliography (RC2):

Dayon, G., 2015. Evolution du cycle hydrologique continental en France au cours des prochaines décennies. Université de Toulouse, Université Toulouse III-Paul Sabatier.

Knutti, R., Furrer, R., Tebaldi, C., Cermak, J., Meehl, G.A., 2010. Challenges in combining projections from multiple climate models. Journal of Climate 23, 2739–2758. doi:10.1175/2009JCLI3361.1

Monerie, P.-A., Sanchez-Gomez, E., Boé, J., 2016. On the range of future Sahel precipitation projections and the selection of a sub-sample of CMIP5 models for impact studies. Climate Dynamics 1–20. doi:10.1007/s00382-016-3236-y

Piazza, M., Boé, J., Terray, L., Pagé, C., Sanchez-Gomez, E., Déqué, M., 2014. Projected 21st century snowfall changes over the French Alps and related uncertainties. Climatic change 122, 583–594.

Quintana-Seguí, P., Le Moigne, P., Durand, Y., Martin, E., Habets, F., Baillon, M., Canellas, C., Franchisteguy, L., Morel, S., 2008. Analysis of Near-Surface Atmospheric Variables: Validation of the SAFRAN Analysis over France. J. Appl. Meteor. Climatol. 47, 92–107. doi:10.1175/2007JAMC1636.1

Quintana-Seguí, P., Turco, M., Herrera, S. and Miguez-Macho, G., 2017. Validation of a new SAFRAN-based gridded precipitation product for Spain and comparisons to Spain02 and ERA-Interim., doi:10.5194/hess-2016-349.

Reifen, C., Toumi, R., 2009. Climate projections: Past performance no guarantee of future skill? Geophysical Research Letters 36. doi:10.1029/2009GL038082

Vidal, J.P., Martin, E., Franchistéguy, L., Baillon, M., Soubeyroux, J.-M., 2010. Uncertainties in the Safran 50-year atmospheric reanalysis over France, in: 11th International Meeting on Statistical Climatology. p. p–39.

Vrac, M., Friederichs, P., 2015. Multivariate—intervariable, spatial, and temporal—bias correction. Journal of Climate 28, 218–237.